# Leaders in the global banking network: Analysis of the Bank for International Settlements network data

**Anthony Bonato**[1]*, **Juan Chavez Palan**[2], **Adam Szava**[1]

**1** Department of Mathematics, Toronto Metropolitan University, Toronto, Ontario, Canada, **2** CIBC Capital Markets, Toronto, Ontario, Canada

* abonato@torontomu.ca

**Data availability statement:** Data on BIS may be found at https://github.com/AdamSzava/BISNetworkData.

## Abstract

Prior work on networks derived from the Bank for International Settlements (or BIS) focused on centrality measures such as degree, betweenness, and DebtRank, with less attention given to adversarial network models. In this work, we address this gap by introducing an adversarial network-based method to locate influential countries within the global banking network. We analyze BIS data from 2000 to 2015, modeling countries as nodes and lending relationships as weighted, directed edges. We study low-key leaders, which are countries with outsize influence despite lower centrality, and highly exposed nodes, which are countries most vulnerable to defaults. Using the Common Out-neighbor (or CON) score with PageRank, we quantify each country's influence and exposure, and define a measure of low-key leader strength. Our results show that low-key leaders, such as those in the United States and Mexico, possess strong influence with lower exposure to contagion, whereas highly exposed leaders, like those in Germany and the United Kingdom after 2003, maintain broad lending portfolios that heighten vulnerability. We also examine these roles over time, including the United States' loss of low-key leader status after the 2008 financial crisis. Our analysis of low-key leaders and highly-exposed nodes provides new insights into systemic risk within the BIS network.

## Introduction

Networks play a central role in analyzing complex real-world systems, ranging from social interactions to financial transactions. In the context of global banking, network analysis offers powerful tools to uncover hidden patterns, assess systemic risks, and develop strategies for regulatory compliance. The interconnected nature of financial institutions implies that vulnerabilities in one part of the system can propagate, resulting in cascading effects throughout the network.

The Bank for International Settlements (or BIS) provides consolidated data on cross-border banking relationships. For more background on BIS, see [1,2]. We can form a network using BIS data, where nodes represent countries and directed edges indicate cross-border lending relationships. The BIS network has been used to model risk contagion and identify

**Funding:** The author(s) received no specific funding for this work.

**Competing interests:** The authors have declared that no competing interests exist.

critical nodes. For a discussion of network methods using BIS data focused on European markets, see [3]. In [4], in- and out-degree distributions were used to analyze consolidated and locational banking statistics in BIS data to measure resilience. In [5,6], degrees and betweenness centrality were used to analyze the global banking network, while [7] used betweenness and eigenvalue centrality. DebtRank, introduced in [8], is a recursive centrality measure used to estimate the impact of shocks in financial networks. DebtRank has been used to study shock propagation and resiliency in banking networks; see [9,10].

Centrality measures such as the Common Out-neighbor (or CON) score and PageRank have been used to detect alliances, leaders, and other influential nodes in adversarial or competition-based network models; see [11–13]. In this study, we analyze the BIS data using the framework of adversarial networks, which represent competitive and antagonistic relationships between nodes. Unlike the earlier studies described in the previous paragraph, we leverage centrality measures such as the CON score and PageRank to uncover two types of influential players: low-key leaders, whose influence is understated by traditional centrality measures, and highly-exposed nodes, which are particularly vulnerable to financial contagion. Our approach using adversarial networks is a novel one and provides insights into systemic vulnerabilities in the global banking system.

## Materials and methods

### Financial contagion in BIS data

The Bank for International Settlements was founded in 1930 to facilitate reparations imposed on Germany after World War I. Reparation tasks became obsolete, and BIS became a provider of statistics and analysis that could aid the cooperation between central banks [2]. Due to globalization over the past century, there have been growing financial interdependencies between developed countries, which have given the BIS a renewed role. The BIS is now heavily involved whenever a financial crisis occurs, as it looks to recommend solutions to preserve financial stability [1]. The institution has expanded to 63 owner countries over the last three decades, becoming a meeting ground for regulators to pursue the standardization of banking practices and demand transparent reporting among the world's major financial players [14].

Since the Latin American debt crisis of 1982, one of the topics of interest at the BIS has been the interdependencies within the *global lending network*; this refers to the worldwide debt held among banks from reporting countries. This dataset is publicly available and published by the BIS Committee on the Global Financial System (or CGFS). While its official purpose is to identify potential sources of stress in the global financial system, it can also be studied to track where a banking crisis in one country can lead to a cascade of defaults in other countries; see [4]. Previous studies conducted by Chen et al. explored financial contagion in the BIS network by sampling only 18 banking systems from this dataset; see [15]. We will extract graphs from the entire global set of 62 CGFS's quarterly data reports from February 2000 to June 2015, modeling them as adversarial networks. Our goal will be to identify the presence of any low-key leaders and to determine if they play a significant role in the network's susceptibility to financial contagion.

The CGFS publishes two sets of international banking statistics every quarter. The first is for *locational statistics*, which is a set of financial statements with assets and liabilities from banks in each reporting country. This information can be significant in the study of a country's internal financial stability. However, as we are concerned with the external interdependencies between countries, we omit locational statistics from our analysis. The second is a set

of *consolidated statistics*, which sums up the accumulated debt from all banks within a country and reports this sum as a consolidated debt-by-country balance. For example, if Canadian banks have lent money through their head office or overseas branches to banks based in France, then the BIS collects the consolidated sum of debt that all French banks owe to all Canadian banks as a single amount. From a financial standpoint, this provides a direct measure of the *risk exposure* each country's banking system has in an external crisis. To continue with our previous example, if the French banking system defaults, all Canadian banks with debt in French bonds would be exposed to the risk of defaulting on their obligations, potentially triggering a crisis in the Canadian banking system.

Three types of consolidated statistics were published in the statistical annex of each BIS Quarterly Review. For our study, we will focus on the *Foreign claims on an immediate borrower basis by BIS reporting country*, which were reported during the 2000 to 2015 period as shown in Table 1. We chose this dataset for its simplicity. The table illustrates the amount of contractual debt held by one country's banking system against another. This was done through their banks' head offices and all their branches and subsidiaries worldwide on a consolidated basis, with any internal bank amounts subtracted. For example, if a bank in Canada has branches in multiple countries, any debt held by those foreign branches or subsidiaries will still be reported as debt held by the Canadian bank. This is because the head office entity in Canada would bear the losses in the case of a default. Any debt held by a Canadian branch located in France is considered internal within the Canadian bank's financial statements. These are not to be counted as part of what debt France would owe Canada. The other two sets of consolidated data have been omitted from our analysis, as they include different attributes for risk mitigation, such as guarantees and collateral, which are tools to prevent the impact of defaults; these would add a higher level of complexity to the interpretations of our model and can be explored in future research.

**Adversarial networks.** *Adversarial networks* are those in which edges model relationships involving competition, dominance, or enmity. We focus here on *competition networks*: if $u$ is in direct competition with $v$, then the direction of the edge $(u, v)$ represents a negative correlation, such as when $u$ owes money to $v$. In [11,12], the authors developed a hypothesis that served as a predictive tool to uncover alliances and leaders within dynamic competition networks, where directed edges are added over discrete time steps.

Consider a dynamic competition network $G$. For nodes $u$, $v$, $w$, we say that $w$ is a *common out-neighbor* of $u$ and $v$ if $(u, w)$ and $(v, w)$ are two directed edges in $G$. Let $CON(u, v)$ be the number of common out-neighbors of two distinct nodes $u$ and $v$. The *Common Out-neighbor*

**Table 1. Global banking network extracted from the BIS Quarterly Review Statistical Annex, June 2002.**

| Country | Austria | Belgium | Canada | Denmark |
|---|---|---|---|---|
| Austria | - | 3,179 | 1,467 | 179 |
| Belgium | 152 | - | 2,080 | 1,291 |
| Canada | 300 | 1,845 | - | 123 |
| Denmark | 349 | 3,194 | 733 | - |
| France | 1,665 | 43,141 | 4,742 | 827 |
| USA | 3,355 | 54,947 | 186,122 | 3,364 |
| UK | 4,191 | 62,365 | 34,328 | 9,781 |

**Table notes:** Nodes represent countries, and directed edges indicate cross-border lending relationships, where an edge from country $u$ to country $v$ signifies that $u$ owes money to $v$. The edge weights correspond to the amount of debt in millions of US dollars.

(or *CON*) score of a fixed node $u$ is defined as

$$\text{CON}(u) = \sum_{v \in V} \text{CON}(u, v).$$ (1)

For a set of nodes $S \subseteq V(G)$ with at least two nodes, we define:

$$\text{CON}(S) = \sum_{u,v \in S} \text{CON}(u, v).$$ (2)

The CON centrality measure has been presented through multiple applications to dynamic competition networks in [11,12]. An influential node is expected to have engagements in sync with many others in the network; conversely, a non-influential node will have limited such interactions. In particular, a high CON score for a fixed node $u$ indicates that it shares many of the same adversaries with other nodes. Hence, $u$ will influence how the network evolves, if even indirectly. A low CON score indicates that a node does not share the same adversaries as other nodes, and therefore, it will not significantly impact the network's evolution. The *Dynamic Competition Hypothesis* (or DCH) provides a quantitative framework for the structure of dynamic competition networks; see [11].

To apply this hypothesis to a banking network, we must first introduce the concepts of an alliance and a leader within a competitive network. *Alliances* are associations formed for mutual benefit, such as countries that pool resources to achieve a common goal. In [12], the authors study social game shows, such as Survivor, where alliances are groups of players who collaborate to vote out players outside the alliance. *Leaders* would be players with a high standing among their peers in the network, and outgoing edges from these leaders will influence edges (which are votes) created by other players. The DCH states that dynamic competition networks arising from an adversarial network satisfy the following properties: (1) Strong alliances have low edge density; (2) members of an alliance with high CON scores are more likely leaders; and (3) Leaders exhibit high closeness, high CON scores, low in-degree, and high out-degree.

The DCH was tested against winners of social game shows, influential actors in political conflicts, and the hierarchical position of biological species in the food chain, as cited in [12]. The authors corroborated that alliances corresponded to nearly independent sets, that CON scores accurately determine the leaders of alliances, and that leaders are detected via their CON scores and closeness.

A contrasting centrality measure for adversarial networks is the well-studied PageRank centrality measure, first introduced in [16]. For additional background on PageRank, see [17]. A high PageRank for a fixed node $u$ in a directed graph will likely correlate with high in-degree nodes. This is why, for an adversarial network, we will compute the PageRank of nodes on a *reversed-edge* network. Therefore, a high PageRank in an adversarial network will correlate with a node that has a high out-degree.

A *low-key leader* (or *LKL*) in an adversarial network is a node whose CON score and PageRank (reversed-edge) are negatively correlated, with a higher CON score and relatively low PageRank. LKLs were first discussed in [13]. Hence, an LKL node would still be influential in the network, but with less centrality than a traditional leader. To be able to calculate a difference between the two values, we note that CON-scores are positive integers and PageRank scores are probabilities, so we re-scale both scores by using *unity-based normalization*:

$$X_{\text{norm}} = \frac{X_i - X_{\min}}{X_{\max} - X_{\min}}.$$ (3)

This scaling measure will satisfy $X_{\text{norm}} \in [0, 1]$ and can create a ranked order of nodes according to either their CON-score or PageRank.

To help illustrate the concept, Fig 1 provides a toy example containing an LKL. In this network, node B has the highest CON score. However, in the reversed-edge network, A has the highest PageRank, and B has a relatively low PageRank, demonstrating that node B is an LKL.

Let $G$ be a directed graph, for the set of nodes $v_i \in V(G)$, where $1 \leq i \leq n$, and the CON-score and PageRank of $v_i$ are denoted by $\text{CON}_i$ and $\text{PR}_i$, respectively, we define *low-key leader strength* as

$$\epsilon_i = \text{CON}_{i,\text{norm}} - \text{PR}_{i,\text{norm}}, \tag{4}$$

where $\epsilon_i \in [-1, 1]$. A node $v_L$ is a low-key leader if it has the maximum value of $\epsilon_L$, and $\epsilon_L > c$, where $c$ is a parameter determined by the network.

In selected sets of adversarial dominance within animal populations, cryptocurrency, and global trade networks, the authors in [13] identified low-key leaders as those with the highest values of $\epsilon$, where $\epsilon > .5$. Furthermore, it was demonstrated that low-key leaders are prevalent in most of the studied adversarial networks.

**Applications to BIS data.** We obtained tables corresponding to BIS Quarterly Review reports from February 2000 to June 2015. See Table 1 for an example of a data extract that shows where each row corresponds to a *debtor* country, whose banking system owes money collectively to foreign *lender* countries' banks. Each column corresponds to the reporting lender country that holds the debt and is owed this amount. We selected sets of adversarial dominance within the 62 adjacency matrices from the datasets. Each node $v \in V(G)$ corresponds to a country, and each weighted directed edge $(u, v; k) \in E(G)$ corresponds to the amount of money $k$ that is owed by country $u$ to the country $v$. The full dataset of adjacency matrices is available on the following GitHub page: https://github.com/AdamSzava/BISNetworkData.

The directed edges $(u, v; k)$ are inherent adversarial relationships, where each debt $k$ from $u$ to $v$ is an adversarial obligation from a debtor to a lender counterparty. Graph $G$ is also a dynamic competition network as it evolves. Each reporting lender country competes with the others to take advantage of new investment opportunities by forming alliances with outlier

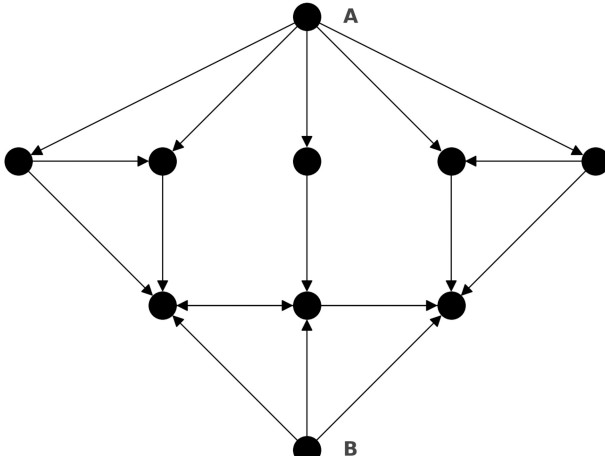

**Fig 1. In the displayed network, node B is an LKL.** Nodes A and B have CON scores of 2 and 8, respectively. In the reversed-edge network, A has the highest PageRank, and B has a relatively low PageRank.

countries. Each debtor country competes with others to obtain additional foreign investment, while each reporting lender country competes to seize new opportunities by forming new alliances. We then compute the weighted CON score and PageRank of each node to calculate the low-key leader strength of all countries across the adjacency matrices.

## Results

### BIS data

Countries with a high-weighted PageRank correlate with nodes that have a high-weighted out-degree. These are countries that owe significant amounts of money to reporting lender countries. As a result of their high centrality, these countries are heavily integrated in the event of a financial contagion and become crucial to the stability of the network. If any of these countries default, then there is a greater risk to the entire network. During the period we studied, Germany, the United Kingdom, and the United States alternated in having the highest PageRank values. Some of the results for March 2006 are shown in Table 2, where we can see that, although the United States owes the largest amount of debt across the network, it does not have the highest PageRank during that same period.

A high PageRank also represents a higher likelihood that the random walk would visit the node at any given time. This has an indirect implication in our adversarial reversed-edge network. Whenever a country falls into default, it is more likely that it would have owed money to the country with the highest PageRank, or through transitivity, it could have owed money to a country that ultimately owes money to the country with the highest PageRank. Consequently, the country with the highest PageRank is at an increased risk of default from any random member of the network.

Countries with a weighted high CON score tend to share similar debt obligations with other nodes; these countries owe large amounts of money to selected lender countries that most other nodes in the network also owe money to. Their debt is both large and diversified, as they share similar debt obligations as their neighboring nodes; hence, the actions of these countries can influence how the network evolves. A low CON score could indicate that a country does not owe a large amount of money and that any significant debt it owes might not be to the same countries that its neighbors owe money to. As seen in Table 3, which compares some results for March 2006, Germany has lent an amount of over $US 3 trillion at this point, which is three times as much as the United States has lent US one trillion, and slightly above the United Kingdom, who are owed US $2.5 trillion. This means that the network as a whole owes more money to Germany than to the United Kingdom or the United States on a consolidated basis, and countries with the highest CON scores are the most likely

**Table 2**. **Countries with the highest weighted PageRank in the BIS banking network for March 2006.**

| Country | PageRank$_{norm}$ | $\sum_u |(v, u; k)|$ |
|---|---|---|
| Germany | 1 | $1,077,765 |
| United Kingdom | 0.9127 | $2,605,395 |
| France | 0.6921 | $735,913 |
| United States | 0.4833 | $4,457,212 |
| Netherlands | 0.4729 | $621,576 |

**Table notes:** The PageRank values are computed on a reversed-edge network, where a higher PageRank indicates greater exposure as a major lender. The total debt column represents the sum of all outstanding loans issued by each country's banking system, measured in millions of US dollars. While the United States has the highest total debt, Germany holds the highest PageRank.

**Table 3.** **Countries with the highest weighted CON score in the BIS banking network for March 2006.**

| Country | $\text{CON}_{\text{norm}}$ | $\sum_{v} |(v, u; k)|$ |
|---|---|---|
| United States | 1 | $1,027,440 |
| United Kingdom | 0.5731 | $2,572,260 |
| Germany | 0.2294 | $3,151,383 |
| Italy | 0.1822 | $360,084 |
| France | 0.1810 | $1,754,414 |

**Table notes:** The CON score measures the extent to which a country shares common debt obligations with others. A high CON score suggests that a country's financial stability is closely tied to that of multiple nations. The total debt column represents the sum of all external liabilities owed to foreign lenders, measured in millions of US dollars.

to owe this money to Germany. A high CON score gives a country the highest influence over causing a financial contagion if it falls into default. It is worth noting that the United States consistently ranks with the highest weighted CON score throughout the entire 2000-2015 period.

**Low-key and highly-exposed leaders in BIS.** For the calculation of the low-key leader strength, we explored maximum values of $\epsilon$ and determined that the absolute maximum $\epsilon = 0.6036$ occurs for the United States in September 2005, as well as the relative minimum and maximum LKL strength values based on interquartile ranges oscillate from $\pm 0.02$ to $\pm 0.08$. Therefore, we determine that a low-key leader for the BIS network exists whenever there is an extremely positive LKL strength outlier such that $\epsilon_L > c$, and $c = 0.1$.

During a brief period from February 2000 to December 2001, a few countries met these criteria, including the United States and the United Kingdom, notably Italy and Ireland. Throughout the period from March 2002 until December 2010, the United States was the only low-key leader that surpassed this threshold. From March 2011 to June 2012, the United States and Mexico shared the status. Eventually, in September 2012, Mexico became the sole leader, a position it held until June 2015. Fig 2 shows the low-key leader strength during March 2006, when the United States was the only low-key leader.

Being a low-key leader in the BIS network correlates with being one of the top lenders of money worldwide, although not necessarily the one with the greatest risk exposure at the very top. They could also have a smaller list of debtor countries than other top lenders. This country is one of the leaders sheltering itself from financial contagion by maintaining a lower level of integration. A relatively higher-weighted CON score ensures that they remain a top player with significant influence over the network's evolution. The low-key leader owes large sums of money compared to other countries with similar PageRank, implying that they are the country that could cause the most damage if they were to default.

An interesting phenomenon also occurs at the other end of the LKL strength graph. Multiple countries are meeting the reversed threshold $\epsilon_L < -c$, as seen in Germany and Japan in Fig 2. This occurrence is not explored in the current literature, so we provide a possible interpretation of the significance of these nodes, which have a high-weighted PageRank and a relatively lower-weighted CON score. In an adversarial network, define a *highly-exposed leader* as one having a much higher PageRank than CON score. Hence, a highly-exposed leader has negative low-key leader strength, typically below some fixed threshold $\epsilon < 0$.

We explored the minimum values of $\epsilon$ and determined that the absolute minimum, $-0.7773$, occurs for Germany in June 2002. While there is a consistent number of countries with $\epsilon$ values below -0.2, only a select few have values below -0.4 across the entire period. Therefore, we determine that a highly-exposed leader for the BIS network exists whenever there is an extremely negative LKL strength outlier such that $\epsilon_L < C$, and $C = -0.4$. Germany

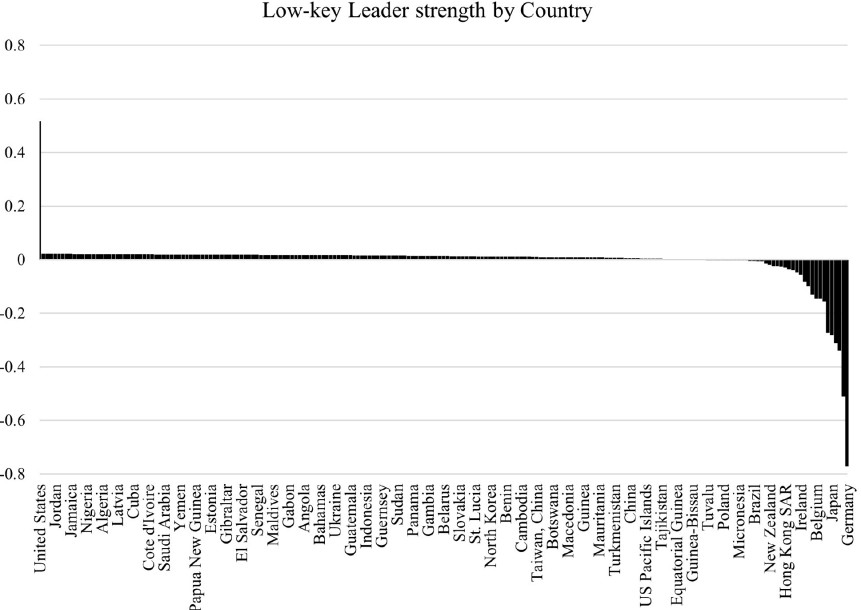

**Fig 2. Low-Key Leader (or LKL) strength in the BIS network for March 2006.** Countries with high positive LKL strength (such as the United States) maintain significant influence while mitigating exposure to financial contagion. In contrast, highly-exposed leaders, such as Germany and Japan, exhibit negative LKL strength, indicating heightened risk due to their extensive lending portfolios.

and France are highly-exposed leaders consistently throughout the timeline. Japan briefly joined during the early 2000's and again after 2013, while the United Kingdom, which initially had high LKL strength, joined the highly-exposed leader group permanently after 2003.

Contrary to the low-key leader, having a negative LKL strength implies that this country has either lent money to the largest list of debtor countries or has the largest amount of money owed to them. This country is, therefore, heavily exposed to the risk of a financial contagion in the network. The relatively lower weighted CON score also means they owe less money than other countries with similar PageRank. Owing less money to some of the major lender countries implies that this country is less influential in the evolution of the network.

We next provide interpretative results for selected countries with high positive low-key leader strengths across the timeline studied, as well as some countries with extremely negative LKL strength values corresponding to highly-exposed leaders. We have also compared our measures of network centrality to a ranking based on a traditional macroeconomic indicator, the expenditure-based nominal *gross domestic product* (or GDP):

$$GDP = C + G + I + NX, \tag{5}$$

where $C$ is consumption, $G$ is government spending, $I$ is investment, and $NX$ stands for net exports [18]. The nominal GDP is a measure of economic output that we can use to rank countries in the BIS network by their productivity. To perform a direct comparison, we performed the same unit normalization on GDP. The result is an evaluation of a country's relevance to the network based on measures of centrality, including LKL strength, CON score, PageRank, and nominal GDP, which provides insight into the actual size of their economies.

**United States.** Most of the major economies are heavily invested in U.S. treasury bonds, as they are considered highly stable [19] by having the highest weighted CON score, the United States has the largest debt in common with its neighbors; this implies that if they were to default first on their debt obligations, they would cause the largest financial contagion to the lending network. They oscillate between other high PageRank countries until the end of 2008, as seen in Fig 3. From around 2009, coinciding with the global financial crisis, they began to increase their integration into the network, eventually overtaking the highest PageRank position by the end of 2012. The United States was our low-key leader from 2000 to 2012, as they were not the most exposed to external risk during that period. Yet, they were always the country that held the most influence, as it exposed the entire network to the most risk if they were to default. While the United States has historically been a key financial hub, studies suggest that it is not immune to cross-border contagion. The 2008 financial crisis demonstrated how financial instability in the U.S. banking sector had widespread global repercussions.

Their LKL strength began to drop after the 2008 global financial crisis, so we can consider a few possible explanations. Following the 2008 financial crisis, investor confidence in the U.S. economy remained strong, as evidenced by the continued high demand for U.S. Treasury securities. While financial markets faced uncertainty, U.S. Treasuries were still viewed as a global haven; see [20]. However, the 52-week U.S. Treasury bond yield dropped below 1.0 percent during the financial crisis and remained low for an extended period, though it fluctuated and began to recover after 2013; see [21]. If we had expected that fewer countries would want to invest in the U.S., we would have seen its weighted CON score decrease. However,

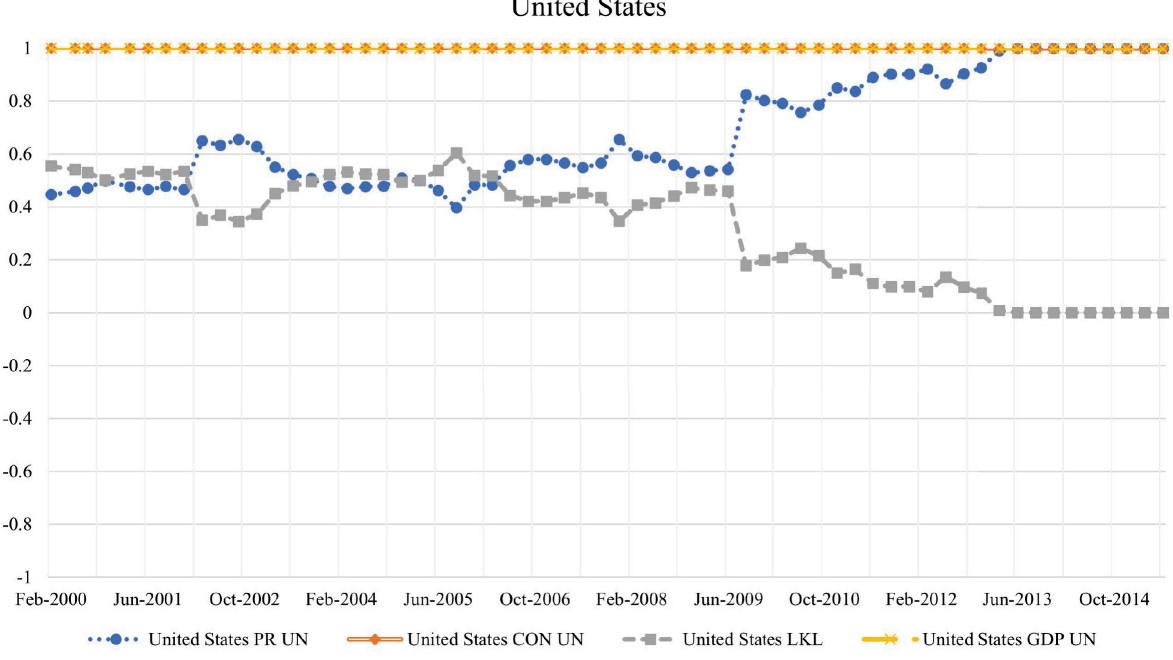

**Fig 3. Evolution of PageRank, CON score, LKL strength, and GDP for the United States in the BIS network.** PageRank (computed on the reversed-edge network) reflects the extent to which the U.S. banking system is a major lender, while the CON score indicates its shared debt obligations with other countries. The LKL strength highlights periods when the United States exerted significant influence with relatively lower direct exposure to contagion risk. GDP is included for comparison, illustrating how financial integration and systemic risk evolved in tandem with economic performance.

by the end of 2012, they had the highest weighted PageRank and the highest weighted CON score. We then consider a hypothesis of re-balancing, since after 2012, the United States is equally exposed to foreign default risk as it is to its own default risk. This re-balancing would align with the fact that they consistently remained the largest economy by GDP throughout the entire period, as seen in Fig 3.

We look for data that can hint towards a large increase in foreign investment; however, the U.S. Bureau of Economic Analysis shows no abrupt increase in direct investment abroad during 2012. To the contrary, it remains steady throughout the whole period; see [22]. Therefore, if the drop in LKL strength is not due to a change in the weighted CON score, then it is solely due to an increase in weighted PageRank. This implies the U.S. has overall increased its exposure to the whole network after 2009, and this would only happen if the re-balancing occurred externally through the overall diversification of risk in the network. This last hypothesis aligns with the following observation: the average LKL strength of all countries with positive LKL values (excluding the US) was 0.016 at the end of 2006, and it later increased to 0.021 by the end of 2014, as more nodes diversified their lending. This is also supported by studies that suggest portfolio re-balancing in European markets was a consequence and a contagion transmitter of the 2008 global financial crisis; see [23].

**Mexico.** In the lead-up to 1982, Brazil and Mexico were two major recipients of large volumes of foreign investment [24]. The domestic banking sector of debtor countries, such as Mexico, was also involved in borrowing from international banking markets. While the BIS monitored international banking activities during the 1980's, the granularity and scope of data collection were not as comprehensive as in later years, potentially limiting detailed insights into the debt volumes and exposures of countries like Mexico during that period. When the Mexican system collapsed in August 1982, known as the Latin American financial crisis of 1982, it triggered a drive for improving policies and reporting for investments in developing countries due to the evident impact on international credit market stability; see [25].

Refer to the values shown for the period 2000-2002 in Fig 4, Mexico has a higher $PageRank_{norm} \approx 0.1$ than $CON_{norm} \approx 0.05$, meaning LKL strength is relatively high in the year 2000 but not enough for Mexico to be deemed a low-key leader. There is a large increase in weighted PageRank during the years 2001-2003, which coincides with the recession experienced by the Mexican economy in 2001-2003 [26], and is claimed to have originated in the U.S. and propagated to the rest of the world [27]. During this time, Mexico's weighted CON score also increased as its total debt in the BIS network went from \$US 62 billion in December 2001 to more than \$US 211 billion in September 2002.

The Mexican economy entered a second recession during the 2008-2009 period [26], and shortly after, the weighted PageRank declined rapidly in March 2010. In 2002, the BIS established a representative office in Mexico City, thereby enhancing collaboration between the BIS and the Bank of Mexico. This facilitated more detailed reporting on international banking activities, including data on debts owed to Mexican banks by other countries' banking systems. The weighted CON score does not change as the debt they owe to other lender countries remains steady from \$US 494 billion in December 2009 to \$US 473 billion in March 2010. The increase in LKL strength may be associated with Mexico's financial interactions with various international financial centers, including jurisdictions like the Cayman Islands, which are known for their significant roles in global finance. Mexico has reduced its risk exposure from external defaults, and the decrease in PageRank is attributed to lending to countries with lower integration in the network. The data show that other countries, such as the United States and Spain, remained heavily invested in Mexico after 2010, and they were exposed to a

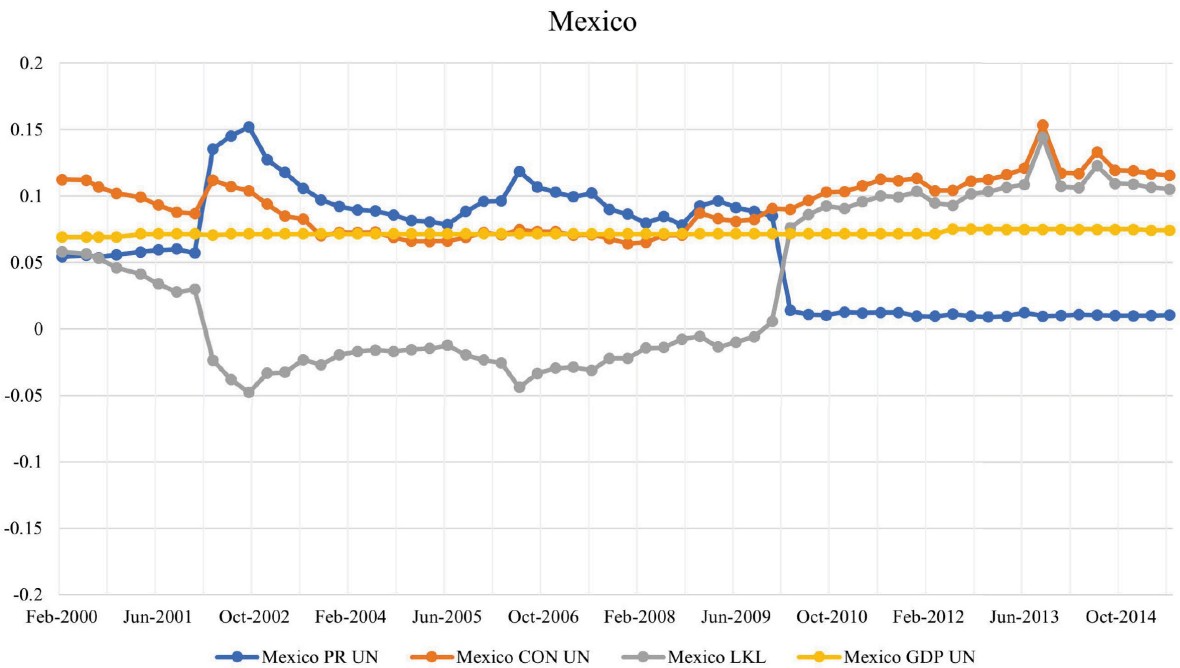

**Fig 4. Evolution of PageRank, CON score, LKL strength, and GDP for Mexico in the BIS network.**

Mexican banking failure because a higher weighted CON score could potentially cause financial contagion. Throughout the final period of 2012-2015, Mexico was the low-key leader of our network by surpassing the 0.1 threshold in September 2012, implying that countries in the BIS network are heavily exposed to a bank default in Mexico, while the sustained lower PageRank implies that Mexico has relatively shielded itself from external default risk.

**United Kingdom.** The United Kingdom is a notable example of a country that transitioned from a low-key leader in 2000 to a highly-exposed leader by mid-2006. As seen in Fig 5, this is due to the large difference between $|\text{CON}_{\text{norm}}| \approx 0.85$ and $|\text{PageRank}_{\text{norm}}| \approx 0.5$; which yields an average low-key leader strength of +0.35 during the first couple of years from 2000 to 2002. The UK has the second-highest weighted CON score throughout the entire timeline, which aligns with some of the conclusions from simulations in [15], which show that the United Kingdom, along with the United States, is one of the most important nodes that can cause a financial contagion in the BIS network.

The weighted CON score decreased slightly after 2002, and the UK became a highly-exposed leader with an LKL strength of -0.45 by 2006. BIS data shows that the amount of debt owed to other countries by UK banks increased slightly from $US 971 billion in December 2001 to $US 1,261 billion in September 2002, while the amount of money that the UK banking system owed doubled from $US 536 billion to $US 1,175 billion respectively. This growth in exposure during the 2005-2014 period can be correlated with the foreign expansion of the UK banking system; see [28]. In 2006, HSBC was among the largest banks in the world based on net assets [29], the year the UK overtook the US and achieved the highest weighted PageRank in the BIS network. The global financial crisis would slow down this growth, along with attempts by the Bank of England and the Financial Conduct Authority to tighten regulations after 2008; see [30].

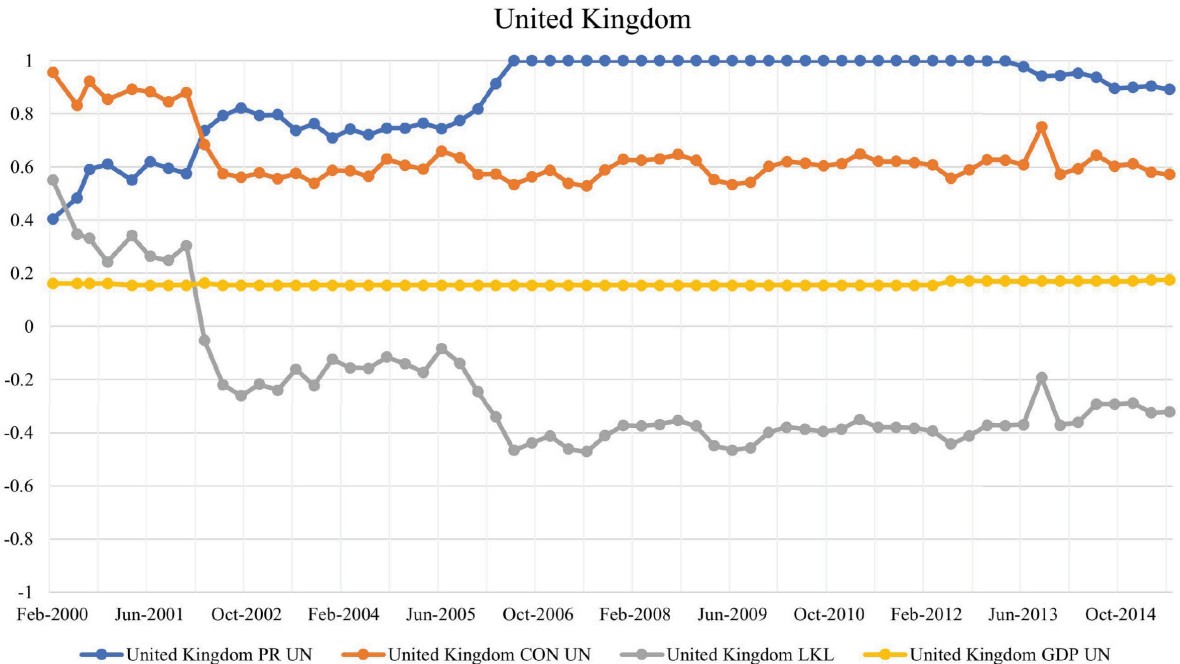

**Fig 5. Evolution of PageRank, CON score, LKL strength, and GDP for the United Kingdom in the BIS network.**

**Italy and Ireland.** Italy and Ireland emerged as low-key leaders during the first two years of the studied timeline; see Figs 6 and 7, respectively. By the end of 2001, they had fallen back below the 0.1 LKL strength threshold. Italy maintained a higher exposure within the financial network leading up to the 2008 financial crisis. This exposure was due to a combination of structural economic challenges and high public debt levels. The collapse of Lehman Brothers in September 2008 further exacerbated these vulnerabilities, leading to a tightening of credit conditions by foreign lenders operating in Italy; see [31]. The post-collapse decrease of credit extended by foreign banks in Italy could show why a wider gap occurs between PageRank and CON score, aligning Italy towards the highly-exposed leaders.

Ireland's PageRank$_{norm}$ and CON$_{norm}$ rankings remained below 0.2, with a minimal difference between them, indicating that Ireland was neither a low-key leader nor a highly-exposed leader during the studied period. However, studies have shown that Ireland's economy is significantly integrated into the global financial system, primarily due to substantial foreign direct investment from multinational corporations; see [32]. This integration suggests that Ireland's exposure to external financial contagion may be more pronounced than some common measures of risk exposure indicate.

**Germany.** Germany has the lowest LKL strength, with values below -0.7 throughout the 2000-2010 period, making it the most highly exposed leader node in the network prior to the 2008 financial crisis. Germany's GDP declined by 4.7% in 2009, reflecting the severe impact of the global financial crisis on its economy; see [33]. It can be seen that Germany has the highest weighted PageRank in Fig 8, which correlates with lending the most amount of money. The characteristic of a highly-exposed leader comprises a relatively lower weighted CON score, implying Germany does not put the network at the greatest risk if it defaults. This is shown in a study conducted in 2022 by Nikkinen et al., as most smaller economies across

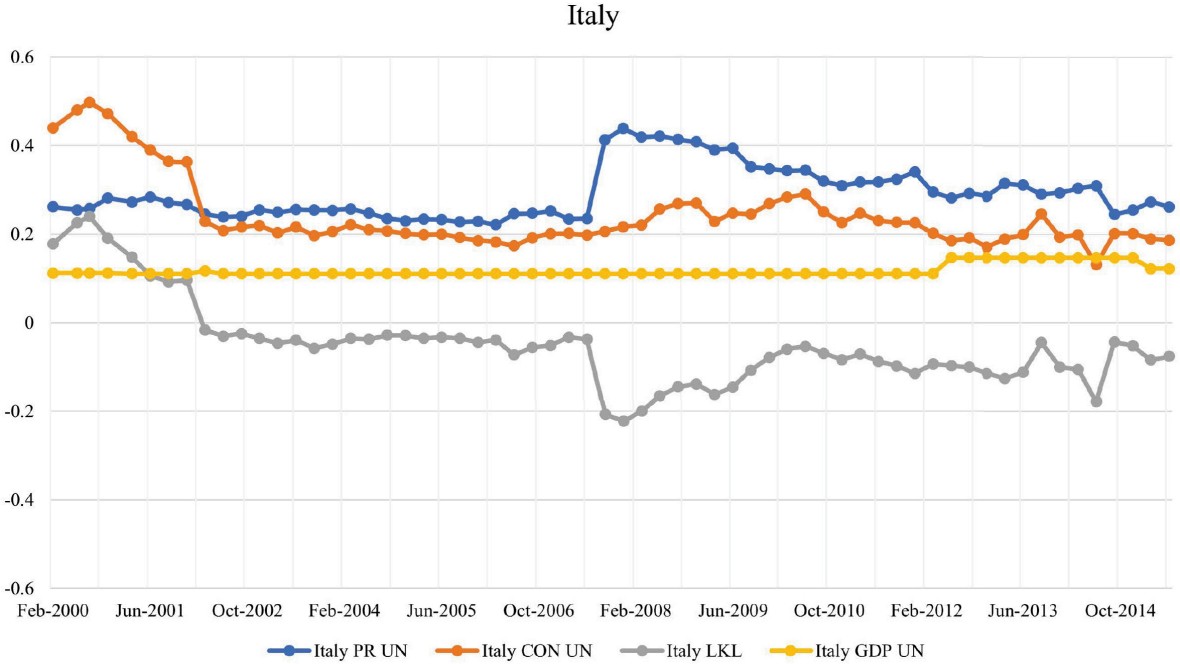

**Fig 6. Evolution of PageRank, CON score, LKL strength, and GDP for Italy in the BIS network.**

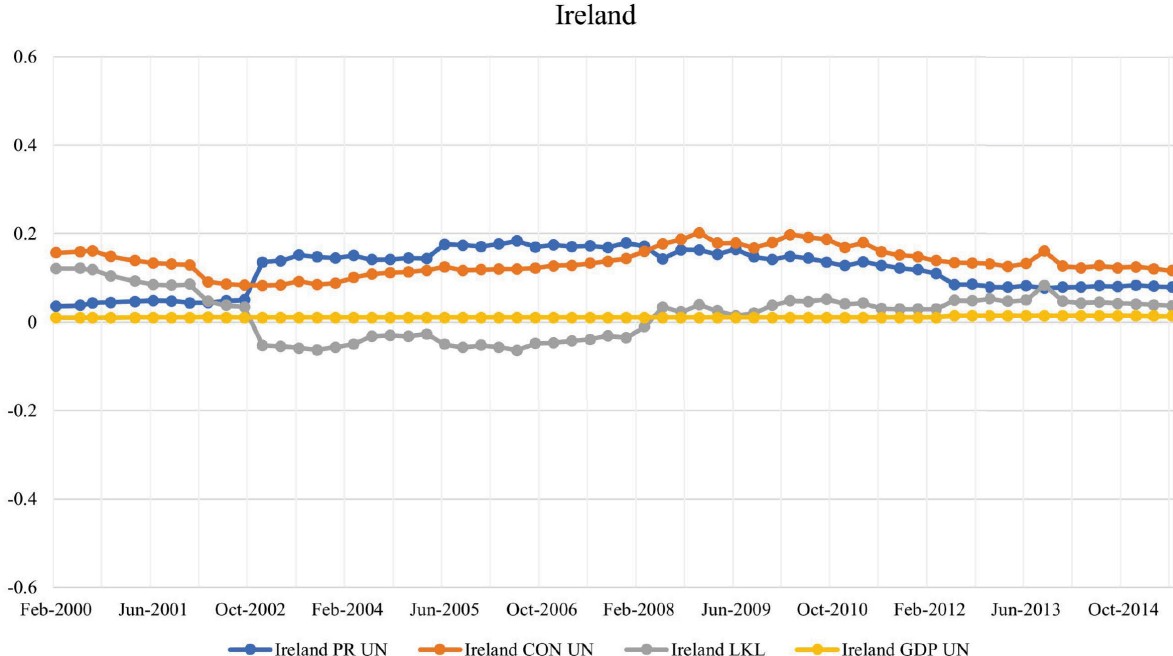

**Fig 7. Evolution of PageRank, CON score, LKL strength, and GDP for Ireland in the BIS network.**

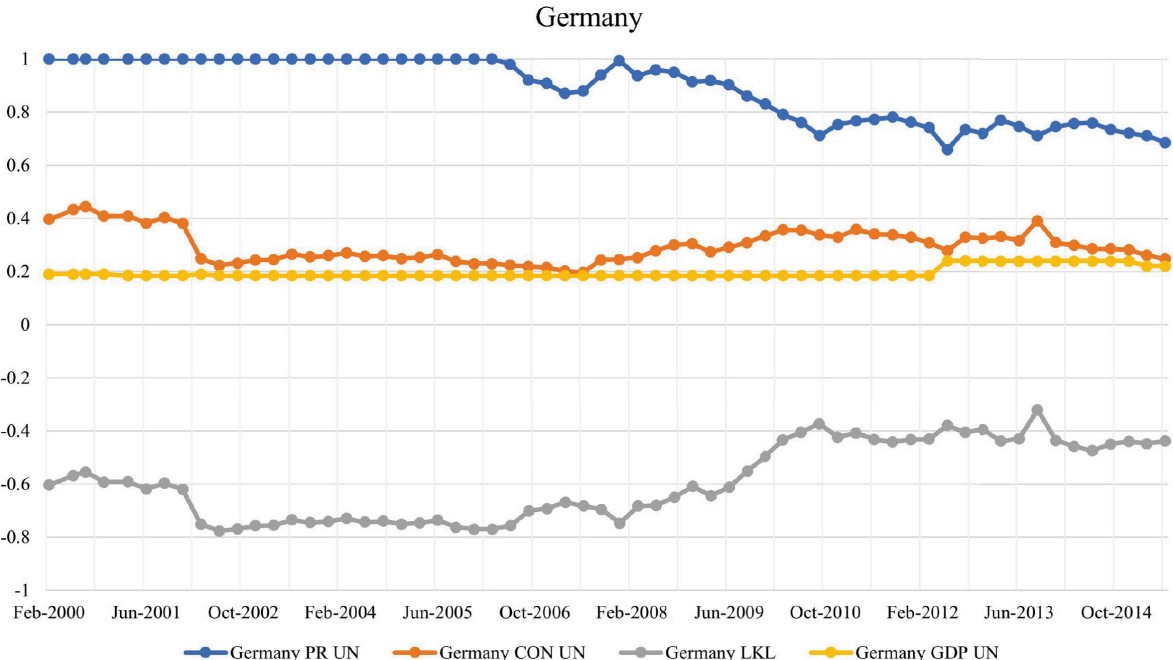

**Fig 8. Evolution of PageRank, CON score, LKL strength, and GDP for Germany in the BIS network.**

Europe were directly affected by the US and not by Germany; only Slovakia's economy was directly affected by Germany at a 5% level. See [34].

After the financial crisis, the value of LKL strength slightly increased to -0.4; while Germany remains a highly-exposed leader, there is a significant decrease in PageRank, which is partially a consequence of the previously discussed overall diversification in the network, which ends up balancing the United States' position to take the highest weighted PageRank by 2012. There is also a significant decrease in risk exposure to external defaults. At the beginning of 2007, Germany's banking system owed $US 4.1 trillion in debt worldwide; by the end of 2012, this number had decreased to $US 2.7 trillion. The retreat is a sentiment felt by investors across Germany as the country tightened its regulations. In response to market conditions, certain German publicly listed companies adjusted their strategies, with some opting to focus more on domestic investors and considering changes to their market listings; see [35].

## Discussion

By analyzing BIS data, we identified low-key leaders and highly-exposed nodes in the global banking network. Our use of adversarial networks and the CON score differs from earlier work that focuses on other centralities, such as degrees, DebtRank, and betweenness.

Our analysis of low-key leaders and highly-exposed nodes provides new insights into systemic risk within the BIS network. We define low-key leaders as countries with low diversification among top lenders. Such leaders subtly expose their peers to the greatest risk yet shield themselves from contagion through lower integration levels. By maintaining a relatively higher weighted CON score, low-key leaders shape the network's evolution. The United States, Mexico, Ireland, and Italy illustrate this phenomenon: they do not provide the highest lending volumes, but, particularly in the case of the United States, they could trigger the largest financial contagion if their banking systems were to default. Finally, the 2008 global financial crisis

affected countries unevenly, and a shift toward greater diversification led the United States to lose its previously low-key leadership status.

We also introduce the new concept of a highly-exposed leader, which is the opposite of a low-key leader and is characterized by a large negative LKL strength value. Such a country carries the greatest risk of contagion within the network. Its relatively low weighted CON score implies it owes less money than countries with similar PageRank values, making it less influential in shaping the network's evolution. Germany and the United Kingdom emerged as prime examples of highly-exposed leaders. Although regulatory bodies in both countries took measures to mitigate risk in the aftermath of the global financial crisis, they still exhibit lower weighted CON scores and relatively higher exposures compared to other leading economies.

Further research into the foreign investment strategies of low-key and highly-exposed leaders may illuminate the policies that enable economies to shield themselves from global financial contagion, particularly when large shifts occur in PageRank or CON score rankings. In future work, it would be interesting to directly link changes in PageRank or CON score to real-world policy decisions.

## Acknowledgments

The authors thank the anonymous referees for their feedback, which greatly improved the paper. The first author acknowledges support from an NSERC Discovery Grant.

## Author contributions

**Conceptualization:** Anthony Bonato, Juan Chavez Palan, Adam Szava.

**Investigation:** Anthony Bonato, Juan Chavez Palan, Adam Szava.

**Methodology:** Anthony Bonato, Juan Chavez Palan, Adam Szava.

**Validation:** Anthony Bonato, Juan Chavez Palan, Adam Szava.

**Writing – original draft:** Anthony Bonato, Juan Chavez Palan, Adam Szava.

**Writing – review & editing:** Anthony Bonato.

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
