## [Decision Letter · Decision Letter 0]

30 May 2025

PONE-D-25-13105Network Analysis of Global Banking Systems and Detection of Suspicious TransactionsPLOS ONE

Dear Dr. Bonato

Thank you for submitting your manuscript to PLOS ONE. After careful consideration, we feel that it has merit but does not fully meet PLOS ONE’s publication criteria as it currently stands. Therefore, we invite you to submit a revised version of the manuscript that addresses the points raised during the review process.

We look forward to receiving your revised manuscript.

Kind regards,

Michel Alexandre, Ph.D.

Academic Editor

PLOS ONE

Journal Requirements:

3. In the online submission form, you indicated that your data is available only on request from a third party. Please note that your Data Availability Statement is currently missing the contact details for the third party, such as an email address or a link to where data requests can be made. Please update your statement with the missing information.

Additional Editor Comments:

As you will read below, both reviewers highlighted the fact that the paper deals with two disjointed subjects. After reading the paper, I strongly agree with their arguments. Thus, I recommend the author to (1) split the work in two, and (2) focus on the first study for the present submission, following the reviewers' suggestions.

Comments from the editorial office: Please review the PLOS policy on the usage of AI tools and technologies

(https://journals.plos.org/plosone/s/ethical-publishing-practice#loc-artificial-intelligence-tools-and-technologies) and provide the following information about your submission.

List all parts of the research and/or submission for which you used generative AI tools, and provide the name(s) of the tool(s) used for each: 

Reviewers' comments:

Reviewer's Responses to Questions

**Comments to the Author**

1. Is the manuscript technically sound, and do the data support the conclusions?

Reviewer #1: Yes

Reviewer #2: Partly

2. Has the statistical analysis been performed appropriately and rigorously? 

Reviewer #1: Yes

Reviewer #2: Yes

3. Have the authors made all data underlying the findings in their manuscript fully available?

Reviewer #1: Yes

Reviewer #2: No

4. Is the manuscript presented in an intelligible fashion and written in standard English?

Reviewer #1: Yes

Reviewer #2: Yes

5. Review Comments to the Author

Reviewer #1: The manuscript, titled "Network Analysis of Global Banking Systems and Detection of Suspicious Transactions" presents a novel approach to understanding the debt dynamics of a global scale banking network by basing the analysis using the Common out-neighbor (CON) score alongside comparisons to PageRank, which was well-fundamented and provides a view into evolving networks anchored to historical interpretations. The authors also presents a cycle detection-based algorithm to identify possible money laundering schemes, which can be a powerful tool to investigators when drawing a list of suspicious transactions.

Although the research has the potential to provide two useful approaches to different scenarios, use cases and applications, I find that the two themes do not have enough in common to be published as a single paper. By being independently of each other, both analysis end up competing to be the main focus of the paper instead of complementing each other.

My suggestion for the authors is to split the paper into two: 1) the analysis for the systemic vulnerabilities of the global banking network and 2) the method for potentially detecting money laundering schemes. I will also add further commentaries to them as two separate studies:

1) The proposal and approach is clear and well defined, with results in the form of figures and tables that suffice to illustrate the method and discuss the main advantages of using both the CON and PageRank metrics to understand the global dynamic of such networks. However, I believe that a brief comparison to other more frequently used network measurements - such as traditional centrality metrics - alongside simulations of attacks and failures of the generated networks and systemic risk indicators (DebtRank and others), given that contagion was discussed and also referenced with previous research, will enrich the paper and help situate the novelty of the proposal to what is currently being used in the literature. Alongside those changes, a larger discussion of how other studies tackled similar problems and what other methods where used (such as the ones mentioned) is essential for future readers to understand the relevance of the proposed approach.

1.1) This study alone is enough for a publication at this journal, should the suggested changes be applied.

2) Differently from the first proposal, I believe that this one still lacks sufficient development to sustain the method proposed. There are many questions still unanswered with the current data and results, such as:

2.1) How we can be sure that the flagged accounts are indeed suspicious (a comparison to annotated data is necessary)? 2.2) What's the effectiveness of the proposed method against others such as supervised methods (data + literature comparison)?

2.3) How would the method perform with another community partitioning algorithm besides Louvain? Also, there lacks and argument in the manuscript as to why it is the best choice.

2.4) Is the path length choice the best one, and why? (The authors argue in the results section that a=4 and b=7 can be good choices for the path length for generating subgraphs inside the detected communities, but no comparison to other choices of parameters were provided)

Overall, in my view, this manuscript can benefit more from focusing solely on the systematic vulnerabilities of banking networks and expanding the respective analysis and discussions, while the second topic can be removed to be a standalone article after further development is done.

Reviewer #2: Essentially this papers describes two different studies. In one study, the authors use an “adversarial network” approach and compare “CON” vs “PageRank” measures to identify “low‑key leaders (LKL) and “highly‑exposed leaders” (HEL) on BIS cross-border exposure data. In the second study, they use Louvain community detection and a “simple-cycles” search to find suspicious transactions for AML applications, in an unsupervised fashion and on a large private Rabobank dataset.

The two studies seem methodologically interesting to me and the BIS study seems largely reproducible since the dataset is made available on GitHub. As such, these studies should be of interest for the interdisciplinary communities working with networks that are readers of PLOS One.

However, reading the paper I found multiple problems that strongly hinder, in my opinion, the publication of this work in the current form.

One main problem lies in the fact that the paper contains two completely separate and disjointed studies. In fact, the two have nothing in common since they are analysis performed with *completely different methods on completely different datasets*. Furthermore, the second study (the one on AML) remains very superficial in terms of the discussion of the results, which seems to me inadequate. A confirmation of this can be found, e.g., in the fact that only a single figure is dedicated to the second study, with no caption and providing really little information.

One potential way to move forward with this paper, in my opinion, would be *dropping the second study*, keeping the results of the second study for a future publication (which should contain a more in-depth analyses), and refocussing this work exclusively on the first study potentially adding further explanations to make it more self contained.

Another important issue is the poor exposition quality of some specific sections that, unfortunately, appear evidently and sadly written by some LLM-based tool. To give a few examples, the sentence “The BIS provides consolidated data on cross-border banking relationships, offering a unique opportunity to study the global banking network” is one of those, given the style and lack of content (in which sense and why would this be a “unique opportunity”? “unique” with respect to what?). Another sentence of this kind, “Our work bridges the study of global banking networks and AML techniques, highlighting the value of network analysis in addressing two critical challenges”, which I think comments on itself.

Frankly, in the age of LLMs, I find it completely unacceptable to have sentences like these published in serious journals. It is disheartening, and it is degrading for the entire profession. As scientists and researchers we should make a serious effort to *resist the temptation* of mindlessly copy-pasting from chatGPT, and actually read what’s written on the pages we are submitting to conferences and journal. In am not suggesting that we shouldn’t use LLMs, but we should always *read carefully* what they write and *rephrase* or *remove* sentences that have no meaning or are badly written. If the authors can’t do this intuitively, I suggest running this tool https://app.gptzero.me/ (or an equivalently good one) on LLM-generated text. Anything below 100% human should not be deemed good enough. Incidentally, I will also suggest the Editor that a similar check should be run on any text that is submitted to the journal.

Minor suggestions.

- The title too generic in my opinion (what kind of network analysis? on which datyaset?) and I think it should be retargeted and made more informative. If the authors follow my suggestion to break the paper in two, this will be easy to do.

- The authors could add an illustrative figure to explain what is a LKL in a practice on a small toy-graph. This would be extremely useful to make the reader understand the methodology immediately, hence also making the paper more self contained.

- The author should probably state name of “Gupta et al algorithm” they use instead of referring generically to the Python routine “simple_cycles" which contains multiple algorithms and might even change over time.

- The authors could clarify why having low CON means having low ability to influence the network, a fact that is given for granted and repeated without sufficient backing in my opinion.

- I think the authors should remove, or rephrase, the sentence "you could say unfairly” as a fairness judgement seems outside the scope for this work.

6. PLOS authors have the option to publish the peer review history of their article (what does this mean?). If published, this will include your full peer review and any attached files.

Reviewer #1: No

Reviewer #2: No

---

## [Author Response · Author response to Decision Letter 1]

12 Jun 2025

Dear Editor,

We want to thank the reviewers for their excellent feedback. Following the advice of both reviewers and the editor, we’ve removed the material on AML, focusing solely on BIS network data. This strengthens the paper and makes it more cohesive. That major change makes the work completely reproducible, as the BIS data is publicly available on our GitHub (while the AML Rabobank data is not public). This change aligns the paper more closely with the mandate of PLOS ONE.

Below is a list of changes made, corresponding to the reviewers' comments. Thanks again for considering our paper.

-Anthony Bonato

Reviewer comment: give brief comparison to other more frequently used network measurements - such as traditional centrality metrics - alongside simulations of attacks and failures of the generated networks and systemic risk indicators (DebtRank and others), given that contagion was discussed and also referenced with previous research, will enrich the paper and help situate the novelty of the proposal to what is currently being used in the literature. Alongside those changes, a larger discussion of how other studies tackled similar problems and what other methods where used (such as the ones mentioned) is essential for future readers to understand the relevance of the proposed approach.

We added a paragraph in the introduction addressing this comment, providing citations to previous work analyzing BIS and other global banking data using network centralities and DebtRank. We ensured that our approach, which focuses on a novel application of adversarial networks, is contrasted with existing methods.

Reviewer comment: Poor exposition in certain sections.

We revised the exposition throughout, especially in the introduction and discussion. We removed the few instances of LLM-generated text, although we relied on tools like Grammarly to polish the manuscript (I think that tool would be acceptable to most scholars). We thank the reviewer for directing us to the GPT Zero site, and we used the basic scan there to evaluate and revise the manuscript.

Reviewer comment: The title too generic in my opinion (what kind of network analysis? on which dataset?) and I think it should be retargeted and made more informative.

Done.

Reviewer comment: The authors could add an illustrative figure to explain what is a LKL in a practice on a small toy-graph. This would be extremely useful to make the reader understand the methodology immediately, hence also making the paper more self contained.

Done. See the new Figure 1, which is a toy-graph that gives an example of an LKL in a small network.

Reviewer comment: The author should probably state name of “Gupta et al algorithm” they use instead of referring generically to the Python routine “simple_cycles" which contains multiple algorithms and might even change over time.

As the AML parts were removed, this was comment no long applies to the current revised paper.

Reviewer comment: The authors could clarify why having low CON means having low ability to influence the network, a fact that is given for granted and repeated without sufficient backing in my opinion.

Done. More discussion was added in the section on Adversarial Networks, especially when introducing the CON score.

Reviewer comment: I think the authors should remove, or rephrase, the sentence "you could say unfairly” as a fairness judgement seems outside the scope for this work.

Done. We reread the paper for analogous fairness judgements to double check and found none.

---

## [Decision Letter · Decision Letter 1]

10 Aug 2025

PONE-D-25-13105R1Leaders in the global banking network: analysis of the Bank for International Settlements network dataPLOS ONE

Dear Dr. Bonato,

Thank you for submitting your manuscript to PLOS ONE. After careful consideration, we feel that it has merit but does not fully meet PLOS ONE’s publication criteria as it currently stands. Therefore, we invite you to submit a revised version of the manuscript that addresses the points raised during the review process.

We look forward to receiving your revised manuscript.

Kind regards,

Michel Alexandre, Ph.D.

Academic Editor

PLOS ONE

Journal Requirements:

Additional Editor Comments:

Request from the Editorial Office: We note that the use of artificial intelligence (AI) tools and technologies were used in the preparation of this manuscript. As per our policy (https://journals.plos.org/plosone/s/ethical-publishing-practice#loc-artificial-intelligence-tools-and-technologies) these contributions must be clearly reported in a dedicated section of the Methods, or in the Acknowledgements section for article types lacking a Methods section. Therefore, in your revised manuscript please include a declaration that states the following information: (1) the name(s) of any tools used, (2) a description of how the authors used the tool(s), (3) how authors evaluated the validity of the tool’s outputs; and (4) a clear statement of which aspects of the study, article contents, data, or supporting files were affected/generated by AI tool usage. Noncompliance with any aspect of this policy will be considered misrepresentation of methods, contributions, and/or results and may result in the rejection of your submission.

Reviewers' comments:

Reviewer's Responses to Questions

**Comments to the Author**

1. If the authors have adequately addressed your comments raised in a previous round of review and you feel that this manuscript is now acceptable for publication, you may indicate that here to bypass the “Comments to the Author” section, enter your conflict of interest statement in the “Confidential to Editor” section, and submit your "Accept" recommendation.

Reviewer #1: All comments have been addressed

Reviewer #2: All comments have been addressed

2. Is the manuscript technically sound, and do the data support the conclusions?

Reviewer #1: Yes

Reviewer #2: Yes

3. Has the statistical analysis been performed appropriately and rigorously? 

Reviewer #1: N/A

Reviewer #2: Yes

4. Have the authors made all data underlying the findings in their manuscript fully available?

Reviewer #1: Yes

Reviewer #2: Yes

5. Is the manuscript presented in an intelligible fashion and written in standard English?

Reviewer #1: Yes

Reviewer #2: Yes

6. Review Comments to the Author

Reviewer #1: The revised manuscript, now entitled "Leaders in the global banking network: analysis of the Bank for International Settlements network data", has addressed all the concerns that I had with its initial version, and seems almost ready for publication.

The only change that I'd recommend for the manuscript would be to rewrite the abstract to be a more descriptive overview of the study performed and the results obtained. Currently, it's too generic and non descriptive enough to be a concise summary of the work contained in the article, and in my view the abstract should be the best opportunity to convince your future readers of the relevance of your research and to read your work.

Try to start it by adding a quick context of the field, with one sentence as the general context and another one tackling the gap in literature that you plan to address. Next, very briefly introduce your methodology, similar to 'In this work, we address these challenges by proposing a methodology that (...)', although this section of the current abstract is clear enough. Also, finish it by highlighting the paper's most relevant findings explicitly.

Reviewer #2: As the authors have addressed the points I raised in my report, I think the article can now be published in PLOS ONE.

7. PLOS authors have the option to publish the peer review history of their article (what does this mean?). If published, this will include your full peer review and any attached files.

Reviewer #1: No

Reviewer #2: No

---

## [Author Response · Author response to Decision Letter 2]

18 Aug 2025

Reviewers' comments:

Reviewer's Responses to Questions

Comments to the Author

1. If the authors have adequately addressed your comments raised in a previous round of review and you feel that this manuscript is now acceptable for publication, you may indicate that here to bypass the “Comments to the Author” section, enter your conflict of interest statement in the “Confidential to Editor” section, and submit your "Accept" recommendation.

Reviewer #1: All comments have been addressed

Reviewer #2: All comments have been addressed

No action taken.

2. Is the manuscript technically sound, and do the data support the conclusions?

Reviewer #1: Yes

Reviewer #2: Yes

No action taken.

3. Has the statistical analysis been performed appropriately and rigorously?

Reviewer #1: N/A

Reviewer #2: Yes

No action taken.

4. Have the authors made all data underlying the findings in their manuscript fully available?

Reviewer #1: Yes

Reviewer #2: Yes

No action taken.

5. Is the manuscript presented in an intelligible fashion and written in standard English?

Reviewer #1: Yes

Reviewer #2: Yes

No action taken.

6. Review Comments to the Author

Reviewer #1: The revised manuscript, now entitled "Leaders in the global banking network: analysis of the Bank for International Settlements network data", has addressed all the concerns that I had with its initial version, and seems almost ready for publication.

The only change that I'd recommend for the manuscript would be to rewrite the abstract to be a more descriptive overview of the study performed and the results obtained. Currently, it's too generic and non descriptive enough to be a concise summary of the work contained in the article, and in my view the abstract should be the best opportunity to convince your future readers of the relevance of your research and to read your work.

Try to start it by adding a quick context of the field, with one sentence as the general context and another one tackling the gap in literature that you plan to address. Next, very briefly introduce your methodology, similar to 'In this work, we address these challenges by proposing a methodology that (...)', although this section of the current abstract is clear enough. Also, finish it by highlighting the paper's most relevant findings explicitly.

A new abstract was included following the reviewer's request. Note that no other further edits were made to the revision.

Reviewer #2: As the authors have addressed the points I raised in my report, I think the article can now be published in PLOS ONE.

No action taken.

7. PLOS authors have the option to publish the peer review history of their article (what does this mean?). If published, this will include your full peer review and any attached files.

Do you want your identity to be public for this peer review? For information about this choice, including consent withdrawal, please see our Privacy Policy.

Reviewer #1: No

Reviewer #2: No

No action taken

---

## [Decision Letter · Decision Letter 2]

13 Sep 2025

PONE-D-25-13105R2Leaders in the global banking network: analysis of the Bank for International Settlements network dataPLOS ONE

Dear Dr. Bonato,

Thank you for submitting your manuscript to PLOS ONE. After careful consideration, we feel that it has merit but does not fully meet PLOS ONE’s publication criteria as it currently stands. Therefore, we invite you to submit a revised version of the manuscript that addresses the points raised during the review process.

We look forward to receiving your revised manuscript.

Kind regards,

Daniel Parkes, PhD

Staff Editor

PLOS One

On behalf of:

Michel Alexandre, Ph.D.

Academic Editor

PLOS ONE

Journal Requirements:

Additional Editor Comments:

In the previous decision you were asked by the Editorial Office to address the below comment:

We note that the use of artificial intelligence (AI) tools and technologies were used in the preparation of this manuscript. As per our policy (https://journals.plos.org/plosone/s/ethical-publishing-practice#loc-artificial-intelligence-tools-and-technologies) these contributions must be clearly reported in a dedicated section of the Methods, or in the Acknowledgements section for article types lacking a Methods section. Therefore, in your revised manuscript please include a declaration that states the following information: (1) the name(s) of any tools used, (2) a description of how the authors used the tool(s), (3) how authors evaluated the validity of the tool’s outputs; and (4) a clear statement of which aspects of the study, article contents, data, or supporting files were affected/generated by AI tool usage. Noncompliance with any aspect of this policy will be considered misrepresentation of methods, contributions, and/or results and may result in the rejection of your submission.

Reviewer's Responses to Questions

**Comments to the Author**

1. If the authors have adequately addressed your comments raised in a previous round of review and you feel that this manuscript is now acceptable for publication, you may indicate that here to bypass the “Comments to the Author” section, enter your conflict of interest statement in the “Confidential to Editor” section, and submit your "Accept" recommendation.

Reviewer #1: All comments have been addressed

2. Is the manuscript technically sound, and do the data support the conclusions?

Reviewer #1: Yes

3. Has the statistical analysis been performed appropriately and rigorously? 

Reviewer #1: N/A

4. Have the authors made all data underlying the findings in their manuscript fully available?

Reviewer #1: Yes

5. Is the manuscript presented in an intelligible fashion and written in standard English?

Reviewer #1: Yes

6. Review Comments to the Author

Reviewer #1: This revision addressed my previous comments concerning the abstract, which is now much clearer and representative of the study. I believe that the manuscript is sound enough to be published at PLOS One.

7. PLOS authors have the option to publish the peer review history of their article (what does this mean?). If published, this will include your full peer review and any attached files.

Reviewer #1: No

---

## [Author Response · Author response to Decision Letter 3]

15 Sep 2025

All responses to the reviewers were addressed in the previous draft. As such, were no additional edits to the manuscript.

---

## [Editor Report · Decision Letter 3]

13 Oct 2025

Leaders in the global banking network: analysis of the Bank for International Settlements network data

PONE-D-25-13105R3

Dear Dr. Bonato,

We’re pleased to inform you that your manuscript has been judged scientifically suitable for publication and will be formally accepted for publication once it meets all outstanding technical requirements.

Kind regards,

Michel Alexandre, Ph.D.

Academic Editor

PLOS ONE
---

## [Editor Report · Acceptance letter]

PONE-D-25-13105R3

PLOS ONE

Dear Dr. Bonato,

I'm pleased to inform you that your manuscript has been deemed suitable for publication in PLOS ONE. Congratulations! Your manuscript is now being handed over to our production team.

Kind regards,

on behalf of

Dr. Michel Alexandre

Academic Editor

PLOS ONE